# OpenReview forum: "Deep Ignorance: Filtering Pretraining Data Builds Tamper-Resistant Safeguards into Open-Weight LLMs"
_ICLR.cc/2026/Conference — ICLR 2026 Poster_

### Official Review · Reviewer_FLB5 · 2025-10-25

**Soundness:** 4
**Presentation:** 3
**Contribution:** 3
**Rating:** 8
**Confidence:** 4

**Summary:**

The authors investigate if removing pretraining data from LLMs can make them more robust to different adversarial attack scenarios (finetuning, evasion-based, etc.). They find that filtering considerably increases robustness in all threat models. Moreover, they analyze limitations of their methods, such as robustness to in-context learning attacks. Overall, this paper presents one of the few works addressing how open-source LLMs can be made safer for public release.

**Strengths:**

- White-box attack settings in open-source models are generally underresearched in the literature. A lot of emphasis is put on the safety of increasingly capable models. How open-source threat models (that are incredibly hard to make save and at the same time very capable) fit in this scenario is mostly ignored.
- The potential safety risk of open-source models is well-motivated
- High rigor (extensive information is provided in the appendix regarding the investigations performed in the paper)
- New approach is compared and combined with the most promising orthogonal methods already investigated in the literature (adversarial training, representation engineering-based training approaches)
- Extensive investigation (and honest reporting) of limitations (e.g., to in-context-learning)
- To the best of my knowledge, the most relevant papers have been referenced. Moreover, the authors specifically discuss their results in context with seemingly conflicting evidence from previous papers.

**Weaknesses:**

Major:
- The takeaway/conclusion of the paper is a bit to optimistic with respect to the experiment results. There are multiple aspects to consider: The academic view: Your approach considerably improves upon the baseline and seems to be a promising direction to explore / an orthogonal defense to many already investigated approaches. Practical view: The results given in Figure 5 show some practical promise in improving the safety of open-source models (harmful data will not always be readily available for fine-tuning).. Yet, finetuning a model for 300m tokens is feasible for basically everyone with more than >=10 dollars, and in most cases, dual-use / harmful data can be scraped from the internet. Thus, currently open-source models do not appear to be safe from misuse at all. In my opinion, this should be more clearly articulated in the paper.

Minor:
- Section 2 provides a lot of very different information in an unstructured way (e.g., one subsection follows after another and addresses very different topics from method design to experiment setup). I believe this could be slightly improved by motivating the structure at the beginning or by clearly separating method design from experiment setup. For example, for somebody not familiar with the literature, the circuit breaker / LAT subsection in 2.3 will be lacking context. Similarly, 2.6 already provides experiment results, mixing three generally separate topics in one section (method, setup, results).
- Summarize the attack threat models in one place (e.g., Latent attacks and Input-space attacks). All of those are evasion attacks.
- The scope of the degradation in "accuracy" evaluation is fitting for this work, but I would assume that a lot of subtle differences/degradations might be difficult to spot, and performing evaluations on a few utility benchmarks might not be sufficient to spot those. A discussion could be added to the paper

**Questions:**

- Do you have an explanation regarding the non-trivial capabilities of the model trained with "strong-filtering" on WMDP, specifically concerning the considerably lower accuracy of the CB+LAT approach (e.g., Figure 3)? I would have expected filtering to be a strictly stronger approach. To phrase it differently, do you think WMDB might not be a suitable benchmark for dangerous capabilities? Based on the results, I would assume that it might be easy to answer some questions of the benchmark in context without any knowledge about the topic (specifically considering the observations provided in Appendix C1)
- The authors acknowledge the limitations of their approach regarding ICL attacks. What exactly do they mean that defending against this kind of threats would require a "defense-in-depth approach". I personally have a hard time coming up with possible defense strategies in these settings.
- Do the authors think that finetuning attacks that try to elicite knowledge with a few training steps might be a relevant threat model in the future against "tamper resistant" models?

---

> ### Author Response · Authors · 2025-11-21
> **Reply part 1: Major concerns**
>
> Thank you for your review and feedback. We are glad that you found the paper to be rigorous and working on a useful problem.
>
> ## 1: Is the paper overclaiming? Is ~10k of tamper resistance enough for meaningful safety?
>
> > The takeaway/conclusion of the paper is a bit to optimistic…finetuning a model for 300m tokens is feasible for basically everyone with more than >=10 dollars, and in most cases, dual-use / harmful data can be scraped from the internet.
>
> We think this is a good observation with important implications for our paper and the broader fields of tamper-resistance and open-weight model safety.
>
> **Where we agree:** In the paper, we work to take tamper resistance up from a few hundred with post-training baselines to about 10,000 with pretraining data filtering. We agree that, in most ways, this is not a huge difference. At the point in which an adversary can do 100 steps of fine-tuning, they already have the compute, code, configuration, and expertise they need to do more. Running fine-tuning for a few thousand steps longer would cost small amounts of marginal money.
>
> **Where we disagree:** However, we think that data may be a key bottleneck. Here, we work with biothreat proxy data using the WMDP bio dataset which consists about 24,000 papers on biothreat proxy subjects. **[Li et al. (2024)](https://arxiv.org/abs/2403.03218) report that making the WMDP bio, chem, and cyber datasets and evals collectively cost over $200,000,** suggesting that collecting high-quality fine-tuning data may be a significant obstacle, costing potentially tens of thousands of dollars and significant expert input. And while our work uses open resources and focuses on biothreat proxy data and evals, it is motivated by safeguarding future systems against potentially more harmful knowledge. It may be substantially harder to obtain the expertise to construct genuinely info-hazardous datasets compared to WMDP-bio, which is a fairly harmless proxy constructed for research purposes.
>
> **The update we are making to the paper:** We are adding a discussion critical of how little compute training on ~10k examples is in absolute terms, and an argument similar to the above for why we believe that data may be the key bottleneck. We are also mentioning that it will be a useful question for future work to study the relationship between the effort required to construct adversarial fine-tuning datasets and their effectiveness.
>
> We also want to thank the reviewer for their reply to Reviewer DY2G and especially for the comments that the quality of the paper is above the median accepted ICLR paper and that "the safety community would be interested in these results and that they should be presented at ICLR."

---

> ### Author Response · Authors · 2025-11-21
> **Reply part 2: minor concerns**
>
> ## 2: Minor
>
> > Section 2 provides a lot of very different information in an unstructured way
>
> Thank you for pointing this out.
>
> Our original goal was to present two results ("Filtering Prevents Target Capabilities" and "Filtering Achieves State-of-the-Art Tamper-Resistance"), one in section 2 and one in section 3. The intuition was that the former would be entirely about filtering and its impacts and that the later would be entirely about tampering attacks. However we agree that the large amount of methodological details necessary in section 2 have obscured our organization. In our updated draft we have sought to clarify the flow of information in section 2 by grouping evaluation details into a subsection of their own, grouping the information that is specific to evaluation practices in a single subsection. We are also adding more signposting language to help readers follow the intended flow.
>
> The new structure is:
>
> 2: Filtering Prevents Target Capabilities
>
> 2.1: Multistage Filtering (details our pipeline)
>
> 2.2: Language Model Training (details our training process)
>
> 2.3: Evaluating Pretraining Filtering (details our evaluation methodology)
>
> 2.3.1: Measuring Biothreat Proxy and General Knowledge (introduces our benchmarks)
>
> 2.3.2: Baseline Post-Training Safeguards (introduces our additional baselines)
>
> 2.3.4: Input-Space Attack (introduces our non-tampering attacks)
>
> 2.4: Results: Filtering is Competitive with State-of-the-Art Post-Training Safeguards (lays out our main claims regarding filtering)
>
> **We will add an explanation of why we chose our baselines in section 2.3.** We chose CB and LAT based on prior work. [Che et al. (2025)](https://arxiv.org/abs/2502.05209) who found in benchmarking experiments that CB was distinctly the strongest post-training method they tested against tampering threats. [Sheshadri et al. (2025)](https://arxiv.org/abs/2407.15549) also found that LAT could combine with and improve over unlearning methods, including to improve tamper resistance. We will also reference Appendix H and Table 9 which audit prior work on tampering with other unlearning algorithms.
>
> > Summarize the attack threat models in one place (e.g., Latent attacks and Input-space attacks).
>
> We believe we already did this in the paragraph titled: “Three tampering attacks: latent-space, adversarial fine-tuning, and benign finetuning.” But if we are misunderstanding something, please let us know.
>
> > The scope of the degradation in "accuracy" evaluation is fitting for this work, but I would assume that a lot of subtle differences/degradations might be difficult to spot
>
> We agree, and troubles with the MCQA evaluations were part of why we reported the *robust MCQA* and *Cloze* evals. **We will add a discussion of the challenges of question-answering accuracy**.
>
> ## 3: Questions
>
> > Do you have an explanation regarding the non-trivial capabilities of the model trained with "strong-filtering" on WMDP, specifically concerning the considerably lower accuracy of the CB+LAT approach (e.g., Figure 3)?
>
> Figure 3 depicts three evaluations, each done with two different question formats, giving six total. The strong filter is very close to random chance accuracy on three of them. So is CB + LAT. While there are plots where CB + LAT has considerably lower accuracy, there are also plots where the strong filter does (namely GCG-U (Cloze)). So we disagree with the implication that the CB+LAT approach is better off the shelf before tampering is performed.
>
> Our intuition is the reverse of yours, that we should expect CB + LAT to be (in some cases) better than the strong filter. The strong filter attempts to remove data from a pretraining dataset that it is not calibrated to and may not contain every way documents in the pretraining corpus talk about the topics in question. There are also multiple hyperparameters to the filter’s design, and we put little effort into optimizing it. CB + LAT, on the other hand, directly optimizes the model to impair its ability to generate representations that correspond to the questions we are evaluating the model on. Intuitively, the latter feels more powerful to us.
>
> Aside from intuition, however, (1) filtering and post-training unlearning algorithms operate on the model through different mechanisms, so we are generally not surprised to see them perform differently, and (2) our goal with figure 3 is to establish that filtering is competing with post-training methods – the key results of the paper relating to tamper resistance follow in section 3.
>
> > What exactly do they mean that defending against this kind of threats would require a "defense-in-depth approach" [against ICL] attacks.
>
> We mentioned defense-in-depth primarily to refer to how the strongest defense we find is from combining filtering and post-training safeguards. That is the only defense resistant to both ICL attacks and tampering attacks. But it’s just not resistant to BOTH at the same time.** We will clarify this.**

---

> > ### Comment · Reviewer_FLB5 · 2025-11-24
> > **Thanks for the response**
> >
> > # 1: Is the paper overclaiming? Is ~10k of tamper resistance enough for meaningful safety?
> > I agree with the data argument. There will be settings where increasing the data threshold will result in a meaningful increase in the effort/cost needed to achieve harm, which ultimately increases safety in practice. I agree with the proposed changes and think they will make the implications of the results clearer to the reader.
> >
> > # Minor
> > Thanks for addressing relevant formatting concerns.
> >
> > > Summarize the attack threat models in one place (e.g., Latent attacks and Input-space attacks).
> >
> > Latent-Space Attacks are not mentioned in 2.3.3. I guess it's not "an input-space" attack, but all attacks follow the common evasion-attack setting, and I would just add latent-space attacks to 2.3.3 (maybe rename it to evasion attacks). However, this is ultimately subjective and does not impact my score.
> >
> > I overall agree with the proposed formatting changes.
> >
> > # Questions
> >
> > Thanks for the explanations!
> >
> > # Score
> >
> > I believe the paper should be accepted to ICLR. Moreover, after reading the other reviews and the rebuttal, it appears that the major concerns have been addressed. I will follow the discussion with the other reviewers and may raise my score depending on their feedback.

---

### Official Review · Reviewer_DY2G · 2025-10-30

**Soundness:** 2
**Presentation:** 1
**Contribution:** 2
**Rating:** 2
**Confidence:** 4

**Summary:**

This paper investigates an innovative approach to enhance the tamper resistance of open-weight models through pretraining data curation. The authors propose a scalable multi-stage data filtering pipeline and pretrain multiple 6.9B-parameter models from scratch based on this methodology.

**Strengths:**

1. The study presents comprehensive experimental validation.

2. The paper is easy to understand.

**Weaknesses:**

1. The paper suffers from disorganized structure, failing to follow the standard methodology-experiments-analysis framework. The presentation appears arbitrary, significantly hindering comprehension.

2. Inadequate baseline comparison: Only Circuit Breaking (CB) and Latent Adversarial Training (LAT) are included. The experimental design should incorporate more baseline methods.

3. Limited dataset evaluation: Sole reliance on the DCLM dataset prevents meaningful assessment of method generalization.

**Questions:**

1. The LaTeX version used by the authors appears inconsistent with the standard ICLR template. For instance, the first-page elements such as "Anonymous authors" and "Paper under double-blind review" are centered in the submitted manuscript, whereas they are left-aligned in the official template. It remains unclear whether other template parameters (e.g., font sizes, spacing adjustments) have been modified.



2. The paper's presentation quality is substantially below acceptable standards, requiring comprehensive revisions to achieve readability.

---

> ### Comment · Reviewer_FLB5 · 2025-11-20
> **Complains feels nitpicked and minor**
>
> I agree with some of the reviewer's concerns on the surface. However, some weaknesses feel very orthogonal to the claim the paper is making and the main results.
>
> First, I would like to point out that the authors present a new and untested approach to get more fundamental safety improvements: data filtering. The experiments in the paper are numerous and require a significant budget. This should be kept in mind.
>
> Weakness 1: I agree to some extent that there are sections that can be improved. However, I would say its above median in quality of ACCEPTED ICLR papers. Moreover, if you have specific issues with the paper, it would be nice to give constructive feedback. Simply stating that it is bad doesn't help anyone.
>
> Weakness 2: The goal is to determine whether data filtering is a viable defense method and to what extent it can aid against model tampering. Experiments should quantify this claim. Comparing the paper to more defensive methods would not improve it in any meaningful way or provide evidence in this direction.
>
> Weakness 3: There are numerous in-depth analyses presented in the appendix. Again, specific suggestions would be constructive.
>
> I strongly believe that the safety community would be interested in these results and that they should be presented at ICLR.
> I would politely ask the AC to check if the concerns about formatting (Question 1) are valid and otherwise strongly consider if this is a meaningful review.

---

> > ### Author Response · Authors · 2025-11-21
> > **Reply part 2**
> >
> > ## 3. The Use of One Dataset
> >
> > > Limited dataset evaluation: Sole reliance on the DCLM dataset prevents meaningful assessment of method generalization
> >
> > **DCLM is a ~500B token pretraining data corpus that is standard in the literature.** [See Li et al. (2024)](https://arxiv.org/abs/2406.11794). We chose DCLM because it was used to train the OLMo2 model [Walsh et al., (2024)](https://arxiv.org/abs/2501.00656).
> >
> > **We cannot think of a reason to expect that our results, which are about filtering biothreat-proxy knowledge from pretraining data, would be qualitatively different for another pretraining dataset, so long as that dataset contained relevant documents related to biothreats.** To be honest, we would be substantially more interested in future work on studying how our methodology interacts with model size and filtering methodology, compared to work that simply varied the training dataset.
> >
> > **Pretraining is very expensive. Rejecting papers that don’t perform runs with multiple pretraining datasets would create significant barriers to work on pretraining science.** Our contract for pretraining compute was for $476K USD. **We are adding this to the paper.** Duplicating our experiments with another pretraining dataset would be very costly. So we believe that it is reasonable to use one pretraining dataset.
> >
> > **Accepting an ICLR paper on pretraining that uses one pretraining dataset is well-precedented.** We used OpenReview to search ICLR 2024 papers for the word “Pretrain” and annotated all papers accepted for oral and spotlight presentations. This turned up 29 papers, of which we felt that 20 were relevant points of comparison for the reviewer’s objection and two were borderline. Two of the papers we labeled as relevant used multiple datasets and one borderline paper did. Across all 22 relevant and borderline relevant papers, the only paper that both did pretraining and used multiple models was [OLMoE: Open Mixture-of-Experts Language Models](https://arxiv.org/abs/2409.02060). However this paper does not use its two datasets in the fashion envisioned by the reviewer. Instead, they show that models trained on their newer dataset were better than models trained on the dataset they used in their last paper. We understand the reviewer’s objection to be “how do we know if the phenomena is real and not an artifact of the dataset you used?” **We believe that that question could reasonably be asked of 22 papers that were accepted for an oral or a spotlight at ICLR 2024 and zero of those 22 papers pretrained their own models on multiple datasets.** There is only one paper where this objection could be reasonably raised and where the authors do as the reviewer requests, but that paper is about finetuning models and pretrains none of the base models that they use themselves. We view this as compelling evidence that the standard the reviewer is holding us to is far outside the norm at ICLR.
> >
> > You can find our annotations [here](https://docs.google.com/spreadsheets/d/e/2PACX-1vQKuHhA9TEq81I9TD5DgZj2R9bRK--l4Byy9Q1Vc51j6Q4T1ibh2PBsGvUjZXi5RBXFJ-mzD1ZY4f7I/pubhtml)
> >
> > ## 3. Centering the authorship block
> >
> > > …the first-page elements such as "Anonymous authors" and "Paper under double-blind review" are centered in the submitted manuscript...It remains unclear whether other template parameters (e.g., font sizes, spacing adjustments) have been modified.
> >
> > Thank you for pointing this out. We used the authblk package to format our names and affiliations nicely. We didn’t notice that this resulted in the author block being centered. We have re-uploaded our submission without this. As best as we can tell, it is formatted exactly according to the ICLR official template.

---

> ### Author Response · Authors · 2025-11-21
> **Reply part 1**
>
> Thank you for your review.
>
> ## 1. Is the paper easy to understand or unreadable?
>
> > The paper is easy to understand.
>
> > The paper's presentation quality is substantially below acceptable standards, requiring comprehensive revisions to achieve readability.
>
> These two claims in the review are directly contradicting. Could you provide some more detail, most importantly, about an aspect or part of the paper that was not clear?
>
> We acknowledge that Reviewer FLB5 felt that Section 2 was a little hard to follow, though they flagged it as a minor concern. We have done a pass on Section 2 based on their feedback.
>
> Meanwhile, Reviewer e68L said the paper was “well written and clear” and that it was easy to find the information they were looking for throughout the paper and appendix.
>
> If you have any specific actionable feedback, we would be happy to work on it. But we are unable to do much with the unsubstantiated comment that the paper was simply unreadable – **both of the other reviews of the paper seem to contradict this.**
>
> ## 2. Baselines.
>
> > Only Circuit Breaking (CB) and Latent Adversarial Training (LAT) are included.
>
> **Why we chose these methods – prior work has established CB and LAT as SOTA baselines:** We chose CB and LAT based on prior work. [Che et al. (2025)](https://arxiv.org/abs/2502.05209) found in benchmarking experiments that CB was distinctly the strongest post-training method they tested against tampering threats. [Sheshadri et al. (2025)](https://arxiv.org/abs/2407.15549) also found that LAT could combine with and improve over unlearning methods, including to improve tamper resistance. **We are moving this explanation from the appendix to section 2.3.**
>
> **Please check footnote 5, Appendix H, and Table 9 for our audit of prior post-training methods:** In those parts of the paper, we explain why we do not use TAR ([Tamirisa et al., 2025](https://arxiv.org/abs/2408.00761)) and how our work has demonstrated a higher degree of tamper resistance than comparable past works on post-training safeguards have tested. **The consensus among prior works is that these post-training safeguards can be undone within dozens or hundreds of steps of adversarial fine-tuning**, e.g. [Che et al. (2025)](https://arxiv.org/abs/2502.05209) and [Qi et al. (2024)](https://arxiv.org/abs/2412.07097). Our paper was written to build upon rather than rehash past works on benchmarking large numbers of unlearning algorithms.

---

### Official Review · Reviewer_e68L · 2025-11-04

**Soundness:** 3
**Presentation:** 4
**Contribution:** 3
**Rating:** 4
**Confidence:** 4

**Summary:**

This paper studies the effect of filtering harmful/dual use data from the training dataset in order to prevent unwanted capabilities. In particular, they propose a simple and automated filtering pipeline using keyword filtering and (optionally) a classifier to remove potential biothreat data from the *pretraining* dataset and train 6.9B-parameter models from scratch. The key findings are that their models maintain utility on standard benchmarks, while having much lower performance on biothreat proxy metrics (WMDP-Bio), and also being fairly resistant to fine-tuning the models on such harmful data. However, they do also find that these ‘filtered’ models are still capable of leveraging the harmful information when provided in context.

First, I think it’s fair to say the results are in line with what would be expected: it’s fairly obvious that training a model with a certain subset of data filtered results in a model that is significantly less performant in that domain. That said, I do believe it is useful to precisely quantify what difference we can expect in practice, so I am generally appreciative of this work. However, I have a few concerns with the results and the framing; I’ll mention my main point below, and mention the rest under the weaknesses/questions sections.

One of the main claims of the paper is “state of the art tamper resistant with respect to fine-tuning”;  I think it’s better to frame this as an empirical study on the relationship between pretraining and fine-tuning data rather than calling this a safeguard. With regards to the adversarial fine-tuning setting, I’m not sure if this is really a good evaluation of whether you are **resistant** to **adversarial** fine-tuning. You fine-tune on `cais/wmdp-bio-forget-corpus`, which contains about ~150M tokens. However we should compare that to the amount of pretraining data filtered: the base model had 300B tokens, with 8.42% filtered (42B), and 50B during the annealing phase with either 9.36% (4.68B) or 4.96% (2.48B) tokens filtered for the strong and weak filter respectively. These are orders of magnitude fewer tokens on these subjects that the filtered model has seen.  I think the missing experiment here is seeing how some addition of the filtered pretraining data would close the gap if incorporated to the fine-tuning phase, as this would be more representative of the “truly worst case” adversarial setting. This is especially relevant because the paper makes the claim that the model is “resistant to adversarial fine-tuning for up to ~305M tokens/10k steps”, while the cloze attacks recover the base LLM’s performance in about 4k (weak filter) and 5.5k (strong filter) steps.

Beyond this, the paper shows that their method is both resistant to input space and latent space attacks attacks (as the knowledge is missing), but also highlights that the lack of knowledge does not prevent the model from using the knowledge if provided in context from other sources. This suggests that additional approaches are needed to mitigate harmful propensities beyond their knowledge-filtering method; the authors show that LAT and CB are effective in this setting. Data filtering thus appears to be a complementary and synergistic component of broader safety strategies.

I currently am leaning towards reject, but I am very open to increasing my score upon discussion and potentially clarifying some of the framing of the paper.

**Strengths:**

- Comprehensive and extensive experiments, including full pre-training of medium scale LLMs, ablations on their data filtration method, comparison with other LLM safety training techniques (LAT/CB).
- Regardless of whether the results are positive, negative, or obvious, large scale empirical studies like this are very valuable to the community
- The paper is well written and clear; there is a lot of information both in terms of intuition as well as technical references to allow users to reproduce their setup if desired. I appreciate how thorough the authors were; most of the times where I had a question about the implementation details, I was able to find it either in the main body of the paper or the appendix.

**Weaknesses:**

- I have several concerns about the evaluations done, in particular the adversarial fine-tuning case (which is one of the main selling points of the work). I have discussed them in the summary section.
- I don’t think I agree with the current framing of how strongly this approach is being sold as a safeguard/robustness technique to adversarial fine-tuning; I see this more as an empirical study on the relationship between the pretraining data and fine-tuning data which is a slight change in the narrative, as well as (in my opinion) missing a few experiments to solidify the results/conclusion.

**Questions:**

- Why are cloze style evaluations marked with the 25% random chance line? My understanding is that this should only apply to the MCQA evaluations?
- Is your adversarial fine-tuning evaluation protocol, which fine-tunes on `cais/wmdp-bio-forget-corpus`, really a fair comparison to what was removed from the base model? What would the results be if you fine-tuned on all the available data, or some interpolation between the regimes?
- How do you think the results would compare with far more powerful/larger pretrained models? Would you expect them to be able to learn harmful data with fewer fine-tuning tokens? If so, would this be a potential issue with scaling this approach?

---

> ### Author Response · Authors · 2025-11-21
> **Reply part 1**
>
> Thank you for your review and feedback. We are glad that you found the contributions “very valuable to the community”.
>
> ## 1: Are the findings obvious?
>
> > I think it’s fair to say the results are in line with what would be expected…
>
> On one hand, we are not surprised that filtering domain data results in less domain knowledge, but we still believe that there is value in demonstrating it and providing models, data, and code for others to experiment with. We also agree with you that it is valuable to “precisely quantify what difference we can expect in practice.”
>
> On the other hand, there is a significant difference between saying “if you remove enough information, you can inhibit these behaviors” and demonstrating a specific methodology that is effective. For example, the fact that such a simple blocklist approach is effective was non-obvious to us and people at NIST who we have presented the research to. We also found our negative results on synthetic document fine-tuning (Appendix J) and our results on filtering for toxicity (Appendix K) to be somewhat surprising. In our view, one of the more interesting contributions of the paper is an understanding that while filtering may lead to tamper-resistance against harmful knowledge in complex domains like bothreat science, it may not lead to tamper-resistance against simpler behaviors like toxicity. See Appendix K for our full discussion.
>
> Finally, after this paper was submitted, a paper was published that cites our preprint and studies the same question in the context of AIs that design biological substances rather than ones that answer questions about biology. They found that the obvious corresponding approach to what we did in their context failed, giving evidence for the idea (raised in Section 6.2) that domain- and modality-specific considerations may influence whether or not pretraining filtering is effective. Given this, demonstrations of success are additionally valuable as they help the community map out the domains in which such methods are successful, which will hopefully lead to a broader understanding of which types of generalizations are and aren’t easy for models.
>
> ## 2: What/how much data should we have adversarially fine-tuned on?
>
> > I’m not sure if this is really a good evaluation of whether you are resistant to adversarial fine-tuning. You fine-tune on…~150M tokens. However we should compare that to the amount of pretraining data filtered…
>
> First, we partially agree. We would certainly hope that removing N tokens of harmful text from the training data would make the model resist N tokens of harmful fine-tuning.
>
> **However, we do not think this is the right goal if we care about practical LLM risk management problems.** Consider the problem from the adversary’s point of view. It does not matter to the attacker how much effort the developer put into risk management – just how easily they can be taken down. For this reason, we feel confident that comparison based on the amount of data required to subvert the defense, rather than efficiency relative to filtered data, is the correct safety-relevant one to make.
>
> **Filtering can be a negative-cost intervention.** Even if we did want to compare the developer’s effort in safeguarding the model versus the adversary’s effort to undo the safeguard, the ratio could be negative! In our case, for the sake of clean comparisons, we added back resampled data so that our filtered models were trained on the same number of tokens as the baseline models. But that wasn’t strictly necessary. The filtering pipeline filtered ~8% of data with an overhead equivalent to roughly 0.8%. So simply omitting it would have been a substantial net savings of effort.
>
> **Our WMDP-bio adversarial fine-tuning evaluation is standard and popular in the literature.** We believe there is much to be said for using comparable attack setups compared to what others are doing in related work. For example, [Wallace et al. (2025)](https://cdn.openai.com/pdf/231bf018-659a-494d-976c-2efdfc72b652/oai_gpt-oss_Model_Safety.pdf) recently took a similar approach, including the same WMDP fine-tuning data in related work.

---

> ### Author Response · Authors · 2025-11-21
> **Reply part 2: the results from the requested experiments**
>
> **We have conducted adversarial fine-tuning on our filtered documents as you described.**
> We have collected 80k documents from our annealing dataset that have among the highest scores according to our ModernBERT classifier and preformed a full-parameter fine-tuning across 150M tokens to see if this is more effective at tampering with our intervention. In this table we report scores at several steps during tampering, with "adversarial" referring to the tampering dataset we originally used  and "filtered" referring to a subset of the filtered documents like the reviewer requested. **The fact that finetuning on the filtered data consistently leads to worse performance than finetuning on the adversarial dataset presents compelling evidence against the hypothesis that the filtered dataset is the best tampering dataset. In fact, our tampering dataset is 3x more sample-efficient than the one the reviewer proposes**. This is consistent for both the baseline and the filtered model, for both evaluation methodologies, and consistently across different checkpoints.
>
> These results are presented in the table below but can be more easily read in Figure 7 (Appendix D) of our updated paper. Note that due to character count constraints some points included in our graph are not included in the table below as the original tables are too large to fit in a message.
>
> **Model = Baseline: No Interventions**
>
> |    Tokens | Experiment                   | Tampering Mix   |   WMDP Bio Robust MCQA |   WMDP Bio Verified Cloze |
> |----------:|:---------------------------|:----------------|-----------------------:|--------------------------:|
> |         0 | Baseline: No Interventions | adversarial     |                 0.432  |                    0.3634 |
> |         0 | Baseline: No Interventions | filtered        |                 0.432  |                    0.3634 |
> |  32768000 | Baseline: No Interventions | adversarial     |                 0.4562 |                    0.3783 |
> |  32768000 | Baseline: No Interventions | filtered        |                 0.424  |                    0.3364 |
> |  65536000 | Baseline: No Interventions | adversarial     |                 0.4389 |                    0.3885 |
> |  65536000 | Baseline: No Interventions | filtered        |                 0.4205 |                    0.355  |
> |  98304000 | Baseline: No Interventions | adversarial     |                 0.4816 |                    0.3903 |
> |  98304000 | Baseline: No Interventions | filtered        |                 0.4228 |                    0.3457 |
> | 131072000 | Baseline: No Interventions | adversarial     |                 0.4793 |                    0.4126 |
> | 131072000 | Baseline: No Interventions | filtered        |                 0.4194 |                    0.3476 |
> | 147456000 | Baseline: No Interventions | adversarial     |                 0.4758 |                    0.4089 |
> | 147456000 | Baseline: No Interventions | filtered        |                 0.4447 |                    0.3541 |
>
> **Model = Data Filtering - Weak Filter**
>
> |    Tokens | Experiment                   | Tampering Mix   |   WMDP Bio Robust MCQA |   WMDP Bio Verified Cloze |
> |----------:|:-----------------------------|:----------------|-----------------------:|--------------------------:|
> |         0 | Data Filtering - Weak Filter | adversarial     |                 0.3422 |                    0.2695 |
> |         0 | Data Filtering - Weak Filter | filtered        |                 0.3422 |                    0.2695 |
> |  32768000 | Data Filtering - Weak Filter | adversarial     |                 0.3721 |                    0.3178 |
> |  32768000 | Data Filtering - Weak Filter | filtered        |                 0.3525 |                    0.2686 |
> |  65536000 | Data Filtering - Weak Filter | adversarial     |                 0.3894 |                    0.3429 |
> |  65536000 | Data Filtering - Weak Filter | filtered        |                 0.3376 |                    0.2825 |
> |  98304000 | Data Filtering - Weak Filter | adversarial     |                 0.3929 |                    0.3615 |
> |  98304000 | Data Filtering - Weak Filter | filtered        |                 0.3548 |                    0.2853 |
> | 131072000 | Data Filtering - Weak Filter | adversarial     |                 0.3836 |                    0.3717 |
> | 131072000 | Data Filtering - Weak Filter | filtered        |                 0.3537 |                    0.29   |
> | 147456000 | Data Filtering - Weak Filter | adversarial     |                 0.4009 |                    0.3773 |
> | 147456000 | Data Filtering - Weak Filter | filtered        |                 0.3652 |                    0.289  |

---

> ### Author Response · Authors · 2025-11-21
> **Reply part 3: On terminology and your other questions.**
>
> ## 3. Should we call it a “safeguard”?
>
> > I don’t think I agree with the current framing of how strongly this approach is being sold as a safeguard/robustness technique
>
> In defense of calling it a “safeguard”:
> Filtering is practical
> It is relevant for safety
> It compares directly and favorably to unlearning baselines
>
> To be honest, your concern seems mostly to be one about word choice. We are not entirely sure why either “safeguard” seems inaccurate or why our method would not be competitive for real-world safety. Could you elaborate a bit more?
>
> ## 4. Questions
>
> **Cloze-style baseline:** For the cloze-style evaluations, we compute the probability of each of the four choices (each with a different query to the model) and select as “the model’s answer” the one that has the highest probability. Details and an example are specified in G.4.2, but we are happy to incorporate feedback on how to make this clearer.
>
> **Adversarial fine-tuning dataset:** See above under item 2.
>
> **Larger models:** This is one of the major outstanding questions from this work (and in some sense, the entire field). We agree this is interesting (it is mentioned in our “future work” paragraph). We speculate that filtering would remain effective at larger scales, but that the sample efficiency of fine-tuning attacks could vary.

---

> ### Comment · Reviewer_e68L · 2025-11-27
> **Thanks for the rebuttal**
>
> **RE: Responses**
>
> Thank you for your responses, I mostly agree with them. I want to emphasize that overall I think the work is good; most of my comments and concerns are with respect to presentation and a few technical details, not that (some) results may be unsurprising. For example, I completely agree with your following comment and am not detracting from the value of your paper by thinking the results are 'obvious'.
> > "On the other hand, there is a significant difference between saying “if you remove enough information, you can inhibit these behaviors” and demonstrating a specific methodology that is effective"
>
>  The two main points I raised that could still be discussed are:
>
> 1. RE: "safeguard/tamper resistance": I can see your perspective on this, but I still think its better to frame this as an empirical study (e.g. on risk mitigation): seeing how much you can push the performance drop on harmful capability evals while minimizing the performance drop on harmless capability evals, and the relationship between seeing this data at pre-training vs fine-tuning. I agree with your comment with respect to practical LLM risk management and have no issue with this (obviously data filtering is important for real-world safety). But my issue with calling it a safeguard is that it seems more like a defence for the model that has the existing capabilities (e.g. safety/refusal training), rather than risk mitigation. For example, [1] is a better example of a safeguard to me, as the tamper-resistance is built in on top of an existing capability, rather than removing the capability altogether. [2] is an example of a blog post written on the subject that which presents it as 'improving safety/mitigating risk' rather than as a safeguard. Ultimately I acknowledge this is a bit subjective, but I really believe the framing I suggest is more appropriate and would strengthen the presentation.
>
> 2. RE: adversarial FT with pre-training data: Thank you for including this experiment. To clarify, I meant to include the filtered pre-training data *in addition* to the WMDP adversarial fine-tuning that you do your primary evaluations on, to account for some of the missing ~40B tokens (the point was if you fine-tune the filtered model on all of the missing data (in the limiting case) and the WMDP documents, how much do you recover?). My understanding is that in these results you only do either the WMDP documents or the filtered documents -- is that correct or did I misunderstand?
>
> **RE: Questions**
>
> Thanks for the clarifications, I think this is good.
>
> **Updated Score: 6**
>
>  Overall, I think this work is good. I appreciate how thorough it is, and believe it should be accepted to ICLR -- I will raise my score accordingly, and will potentially raise my score further upon clarifying/discussing the previous points.
>
> NOTE: I am currently unable to raise my score on my official review (perhaps tied to the recent bug compromising user privacy). My current score for this paper is 6, NOT 4 as in my original review.
>
> [1] Tamirisa, Rishub, et al. "Tamper-resistant safeguards for open-weight llms." arXiv preprint arXiv:2408.00761 (2024).
>
> [2] Chen, Yanda, et al. "Enhancing model safety through pretraining data filtering." Anthropic Alignment Blog (2025).

---

> > ### Author Response · Authors · 2025-12-01
> > **Thanks!**
> >
> > Thanks for the reply!
> >
> > ### 1. Safeguards/tamper resistance
> >
> > **Thanks -- we see your perspective too, and we largely agree.** We view building systems from the ground up to have safety properties a safety intervention **on the process that produces the model.** You are right to point out that there is no unsafe model to safeguard in our set-up.
> >
> > **In defense of calling it a safeguard:** Despite the above, we are nevertheless intervening on a process that has a tendency to produce unsafe things to make it produce more safe things. So we are putting a safeguard on that **process** not on the model it produces. We also think that there is an advantage to calling it a "safeguard" in the title. It helps to communicate the purpose of tamper resistance in this context. Readers who don't work on this exact thing might read a title like "Deep ignorance, filtering training data builds *tamper resistance* into open-weight models" and not know that it's a paper about managing risks.
> >
> > **We'd consider updating the "safeguard" terminology, but we have a bias against it due to a public preprint of the paper.** In our experience, title updates tends to cause some complexity/messiness when searching and citing papers.
> >
> > **Question:** Could you give a little bit more detail about what changes you would envision to the paper. For example, would it be possible to address your thoughts in full by reworking all sentences of the paper (including the title) that call filtering a safeguard or something equivalent?
> >
> > ### 2. Safeguards/tamper resistance
> >
> > **On one hand, if we mixed datasets in our experiments, we would expect the results to be an interpolation of our previous results.**
> >
> > **But we see your main point, and we agree.** We think that it is interesting how a model's domain capabilities vary as a function of how domain data was distributed throughout the pretraining process. Past work like [Feng et al. (2024)](https://arxiv.org/abs/2412.15285) has shown that data ordering can make a big difference to a model. We also know from papers like [Susnjak et al. (2025)](https://dl.acm.org/doi/full/10.1145/3715964) that taking a pretrained model and then fine-tuning it on a lot of domain-specific data is effective for increasing its capabilities in that domain. But we do not know of any papers that have directly studied the question you are raising. We will add this to the discussion.
> >
> > Thanks again for the reply and for updating your review!

---

### Meta-Review · Area_Chair_wAge · 2026-01-11

**Summary:**

The submission underwent good amount of discussion between the authors and the reviewers. After the discussion the paper receives overall positive ratings (6 and 8) except one less-engaged reviewer's score (2). On the two actively engaged reviewers-author discussion threads, the AC found that the submission is more than showing an empirical results of intuitively straightforward idea that filtering some data would enfore robustness to different adversarial attacks. There are multiple discussion about whether the method should be called as ""safeguards"" or not and the empirical results may be trivially tried using marginal amount of money. But these issues are well addressed by the authors' responses and the reviewers agree to the authors' arguments and raise the score to acceptance side. While reviewer DY2G raised some concerns about presentation quality, the AC also agree that some of the presentation should be improved (will list them at the end of the meta review) but thinks that it is not a major issue for the submission to be rejected but to be revised for the final copy of the paper. Although the initial rating of the submission was not compelling for acceptance, the discussion makes the submission to be worthwhile to be reported to be community, the AC thinks. So, the AC recommends the paper to be accepted in ICLR 2026 with the reservation of revising the following presentation items.

- The font size of some figures (e.g., Fig 1,2,6) should be in the size of the caption font size. Otherwise, it looks not coherent to the text and the caption. I recommend to reduce the size of the fonts (especially the title of the plots) to be smaller and similar to the caption font-size.
- Left margins of bullet point list or item list should be reduced.
- Paper should fit to 9 pages. Reduce the text to fit to the page limits.
- Related work section would be better in the main paper. You can reduce the arguments (or briefly introduce some of the important ones) in the related work in the main paper (say 1/3 of page) and point detailed discussion to the elaborated version of related work section in the appendix.

**Reviewer Concerns:**

Some of the DY2G's comments are worth to review again. All other reviewers' concerns seems well addressed during the author-reviewer discussion phase.

**Reviewer Scores:**

DY2G needs to join the discussion with the authors and reviewer FLB5 for more convincing ratings and feedback.

---

### Decision · Program_Chairs · 2026-01-26

Accept (Poster)